# Physiological, Anatomical, and Agronomic Responses of *Cucurbita pepo* to Exogenously Sprayed Potassium Silicate at Different Concentrations under Varying Water Regimes

Enas S. Azab [1],*, Khalid S. Alshallash [2],*, Mesfer M. Alqahtani [3], Fatmah A. Safhi [4], Salha M. ALshamrani [5], Mohamed A. M. Ali [6], Taia A. Abd El-Mageed [7] and Ahmed M. El-Taher [8]

[1] Agricultural Botany Department, Faculty of Agriculture, Suez Canal University, Ismailia 41522, Egypt
[2] College of Science and Humanities-Huraymila, Imam Mohammed Bin Saud Islamic University (IMSIU), Riyadh 11432, Saudi Arabia
[3] Department of Biological Sciences, Faculty of Science and Humanities, Shaqra University, Ad-Dawadimi 11911, Saudi Arabia
[4] Department of Biology, College of Science, Princess Nourah bint Abdulrahman University, Riyadh 11671, Saudi Arabia
[5] Department of Biology, College of Science, University of Jeddah, Jeddah 21959, Saudi Arabia
[6] Department of Horticulture, Faculty of Agriculture, New Valley University, El-Kharga 72511, Egypt
[7] Soil and Water Department, Faculty of Agriculture, Fayoum University, Fayoum 63514, Egypt
[8] Agricultural Botany Department, Faculty of Agriculture, Al-Azhar University, Cairo 11651, Egypt
* Correspondence: enassafaa2010@hotmail.com (E.S.A.); ksalshallash@imamu.edu.sa (K.S.A.)

**Abstract:** Drought is one of the major environmental stresses that devastatingly impact squash development, growth, and productivity. Potassium silicate can attenuate the injuries caused by water stress. Hence, this study was designed to investigate the influence of three concentrations of potassium silicate; 10, 15, and 20 g/L on squash plants versus untreated control under three irrigation regimes; 100, 75, and 50% of estimated crop evapotranspiration (ET). The obtained results indicated that moderate (75% ET) or severe (50% ET) drought stress conditions gradually declined photosynthetic pigments, relative water content (RWC), mineral content, physiological parameters, and anatomical characteristics. These deleterious impacts were reflected on all growth and yield traits, i.e., plant height, fresh and dry weight of root and shoot, and fruit yield. On the other hand, the antioxidant enzyme activities; superoxide dismutase (SOD), catalase (CAT), and peroxidase (POX) significantly increased under severe drought stress at 50% ET followed by 75% ET. However, all evaluated exogenous applications of potassium silicate substantially enhanced photosynthetic pigments, RWC, N, P, and K content, antioxidant enzyme activities, and anatomical characters (periderm thickness, cortex thickness, midrib thickness, mesophyll thickness, number of xylem vessels per main vascular bundle, thickness of vascular bundle, thickness of collenchymatous tissue and upper epidermis, and thickness of collenchymatous tissue and lower epidermis). These desirable impacts were reflected in enhancing all growth and yield parameters. Conclusively, this study alludes that the exogenously applied of potassium silicate, particularly at 20 g/L, can alleviate the deleterious effects of drought stress and enhance the growth and productivity of squash plants, especially in arid environments.

**Keywords:** drought stress; photosynthetic pigments; antioxidant enzyme activities; fruit yield

## 1. Introduction

Squash (*Cucurbita pepo* L.) is one of the most popular cultivated vegetable crops [1]. The immature fruits are used as boiled, cooked, or stuffed [2]. It has various health and medicinal benefits for humans [3,4]. Its fruits contain considerable amounts of carbohydrates, proteins, minerals, and vitamins [5,6].

Global climate change is causing considerable fluctuations in various variables, particularly temperature and precipitation [7,8]. Such shifts contribute intrinsically to environmental stress exposure in plants [9–12]. Drought is one of the major abiotic stresses and the most considerable factor limiting plant growth and productivity, in particular in arid environments [13–15]. It destructively influences anatomical, physiological, and morphological characteristics of arable field crops and subsequently food security [16–19]. The devastating impacts of drought stress on the performance of different field crops have been broadly investigated [20–24].

There are certain non-essential elements, such as silicon (Si), that improve plant development and growth through promoting several desirable physiological processes [25,26]. Furthermore, silicon alleviates destructive impacts of different abiotic stresses, such as nutrient imbalance, salinity, drought, metal toxicity, chilling, radiation damage, and high temperature [27,28]. Silicon increases tolerance to drought stress by sustaining plant water balance and photosynthetic activity [29,30]. The valuable role of Si in promoting the development and growth of different plant species under drought stress has been detected in previous studies. In this regard, Moustafa [31] demonstrated that exogenously applied potassium silicate significantly enhanced growth characters, pigment content, and N, P, and K absorption in *Ceiba speciosa*. Likewise, Kaya et al. [32] deduced that the application of Si enhanced relative water content in maize (*Zea mays*) under water deficit conditions, indicating retention of water in cells. Moreover, Kaya et al. [32] disclosed that Si has a vital role in the regulation of calcium concentration which strengthens plant membrane integrity under different abiotic stresses. Pei et al. [33] manifested that Si application stimulated antioxidant defense in wheat (*Triticum aestivum*) produced under water deficit conditions. Consequently, the antioxidant defense alleviated oxidative damage induced by the overproduction of reactive oxygen species (ROS) and maintained physiological processes in stressed plants. Thus, the purpose of this study was to explore the effect of K-silicate as a foliar spray at different concentrations under different irrigation regimes on physiological parameters, anatomical characteristics, and growth and yield traits of the squash plant (*Cucurbita pepo* L.).

## 2. Materials and Methods

### 2.1. Experimental Site and Agronomic Practices

A field experiment was performed during the two growing summer seasons of 2019 and 2020 at the Experimental Farm of the Faculty of Agriculture, Suez Canal University, Ismailia, Egypt (30°35′45″ N, 32°16′18″ E). The soil of the experimental site was sandy throughout the profile (92.1% sand, 7.52% silt, and 0.47% clay), the electrical conductivity and pH were 1.4 dSm$^{-1}$ and 7.41, respectively. The experimental site is described as arid with no precipitation occurring during the summer season (Table S1). The applied experimental design was a split-plot in three replications. The irrigation regimes were randomized in the main plots and the foliar application of potassium silicate was located in the subplots. Irrigation scheduling was based on the estimated crop evapotranspiration (ETc) replacement according to the crop coefficient approach of Allen et al. [34]. ETc was calculated by multiplying the daily reference evapotranspiration (ETo) by FAO crop coefficients (Kc) of squash [34]. Daily reference evapotranspiration (ETo) was calculated from weather data using the FAO-56 standardized Penman–Monteith equation as stated in [34]. Daily meteorological data, including maximum and minimum temperature, wind speed, and dew point temperature, were obtained from a weather station located at the experimental site and were used for calculating ETo. The values of Kc for squash, as recommended by FAO-56, were adjusted based on actual values of climatic conditions, including relative humidity and wind speed in the experimental site. Three irrigation regimes were performed using 100%, 75%, and 50% of ETc. The total amount of the full irrigation regime (100% ETc) was 2105 and 2195 m$^3$/ha during the first and second growing seasons, respectively. The amount of full irrigation regime was diminished by 25% for providing moderate drought stress conditions, which were 1579 and 1646 m$^3$/during first and second growing seasons, respectively. Besides, the amount of full irrigation regime was

diminished by 50% for providing severe drought stress conditions, which were 1053 and 1098 $m^3$/ha during the first and second growing seasons, respectively. The drip irrigation system was applied, and the emitter flow rate was 4 L $h^{-1}$. The irrigation water amount was calculated independently for each irrigation regime employing a flow meter. The irrigation regimes were applied after two weeks from planting to ensure full complete seedling establishment.

Potassium silicate $K_2SiO_3$ 42% was purchased from Sigma and was applied in three concentrations of 10, 15, and 20 g/L. The plants were sprayed three times starting from the age of twenty days with one-week intervals among sprayings. The experimental subplot consisted of 5 rides, 5 m in length and 60 cm in width. The squash (*Cucurbita pepo* L.) zucchini type cv "Eskandarani" was used for this study. According to the recommended period of squash growing in the region, planting was performed in mid-May in both years. Standard agronomic practices, including drip irrigation and control of pests and diseases, were performed as recommended for cultivating squash in the region. Before sowing, 360 kg super-phosphate (15.5% $P_2O_5$) per ha and 240 kg potassium sulfate (48% $K_2O$) were applied. Additionally, 240 kg ammonium sulfate (20.5% N) was added.

*2.2. Physiological Parameters*

The contents of photosynthesis pigments *Chl a*, *Chl b*, and carotenoids were determined in the 3rd leaf of squash after 50 days from sowing according to Von Wettstein [35]. Leaf relative water content (RWC) was estimated as outlined by Schonfeld et al. [36]. Fresh leaf samples (100 mg) were soaked in 10 mL of distilled water until saturated and left overnight. After removing water from the leaf surface without pressure, the leaf was dried at 70 °C for 72 h and weighed to obtain a saturated weight. The dry weight was obtained. From these data, RWC was calculated using the following equation RWC = (Fresh weight − dry weight)/(Turgid weight − dry weight) × 100.

The terminal buds and the first two young leaves were utilized for determining the activities of superoxide dismutase (SOD), catalase (CAT) peroxidase (POX), and polyphenol oxidase (PPO) enzymes. Two grams of the plant material were homogenized with 10 mL of phosphate buffer pH 6.8 (0.1 M) and centrifuged at 2 °C for 20 min at 20,000 rpm. The clear supernatant (containing the enzymes) was taken as the enzyme source [37]. Superoxide dismutase (SOD) activity was determined following Marklund and Marklund [38] and Kong et al. [39]. The solution (10 ml) consisted of 3.6 ml of distilled water, 0.1 mL of the enzyme, 5.5 ml of 50 mM phosphate buffer (PH 7.8), and 0.8 mL of 3 mM pyrogallol (dissolved in 10 mM HCl). The rate of pyrogallol reduction was measured at 325 nm with UV-spectrophotometer. One unit of enzyme activity was defined as the amount of the enzyme that resulted in 50% inhibition of the auto-oxidation rate of pyrogallol at 25 °C. Catalase (CAT) activity was determined according to Chen et al. [40] and Kong et al. [39]. CAT activity was determined by measuring the rate change of $H_2O_2$ absorbance in 60 s with a UV-spectrophotometer at 250 nm. The blank sample was made by using buffer instead of enzyme extract. One unit of enzyme activity was defined as the amount of the enzyme that reduced 50% of the $H_2O_2$ in 60 s at 25 °C. Peroxidase (POD) activities was assessed according to Bergmeyer and Bernt [41]. The enzyme was assayed using guaiacol as the substrate. The reaction mixture consisted of 3 mL of phosphate buffer (0.1 M, pH 7.0), 30 mL of $H_2O_2$ (20 mM), 50 mL of enzyme extract, and 50 mL of guaiacol (20 mM). The reaction mixture was incubated in a cuvette for 10 min at room temperature. The optical density was measured at 436 nm and the enzyme activity was expressed as the number of absorbance units $g^{-1}$ fresh weight of leaves. Polyphenoloxidase (PPO) (s) activity was determined according to Kar and Mishra [42] and Fick and Qualset [43]. PPO activity was assessed by using 125 μmol of phosphate buffer (pH 6.8), 100 μmol pyrogallols, and 2 mL of enzyme extract. After the incubation period of 5 min at 25 °C, the reaction was stopped by adding 1 mL 5% $H_2SO_4$. The blank sample was made by utilizing very well-boiled enzyme extract, and the developed color was read at 430 nm.

After 50 days of sowing, leaves and roots of squash plants were collected and dried at 70 °C for 24 h. Analyses of nitrogen, phosphorus, and potassium were performed by grinding the dried plantlets followed by digestion with $H_2SO_4$ as described by Piper [44]. The nitrogen, phosphorus, and potassium contents were then measured by using an Atomic Absorption flame photometric (3300) according to Wild et al. [45].

For nitrogen, digested samples were diluted, neutralized with sodium hydroxide, and analyzed with an ammonia electrode. Ammonia concentration was estimated from a standard curve prepared from serially diluted standards of a 1000 mg/L ammonium chloride stock solution [46]. Phosphorus was determined according to Jackson [47].

### 2.3. Anatomical Characters

Certain characteristics of transverse sections of the third visible leaf from the plant apex were determined, such as the thickness of mesophyll, thickness of midrib, thickness of vascular bundle, thickness of collenchaymatous tissue and upper epidermis, thickness of collenchymatous tissue and lower epidermis, and the average number of xylem vessels per vascular bundle. Fixation of leaf sample in 70% formalin acetic acid (F.A.A.) solution, dehydration and clearing with ethyl-alcohol and xylene, infiltration and embedding in pure paraffine wax (M.P. 56 to 58 °C) were performed. Utilizing a rotary microtome, sections of the leaf (15µ) were stained with safranin and light green. The sections were investigated microscopically using the image processing program Image. The anatomical assessment was performed employing a Leica light Research Microscope model PN: DM 500/13613210 supplied with a digital camera [48].

### 2.4. Growth Traits

At 50 days after sowing, ten plants were randomly selected to measure root length (cm), fresh and dry weights of shoot and root per plant (g), plant height (cm), number of leaves per plant, and fruit weight per plant (g).

### 2.5. Statistical Analysis

The R statistical software (version 4.1.2) was employed to analyze the obtained data. A combined analysis of variance was performed for the split-plot design with irrigation regime, potassium silicate application, and their interaction. Combined analysis of variance was performed to explore the differences among studied factors across the two growing seasons using the Shapiro–Wilk test and Bartlett's test for the normality distribution of the residuals and homogeneity of variances, respectively. The combined analysis indicated homogenous variances across the two growing seasons for different parametric measurements, and therefore, the data of the two growing seasons were combined. The differences among studied factors and their interaction were separated by the Tukey HSD test at 0.05 significance level.

## 3. Results and Discussion

### 3.1. Physiological Parameters

3.1.1. Relative Water Content and Photosynthetic Pigments

Relative water content (RWC) was determined to provide an indication of the plant water status affected by the studied factors. The results in Figure 1 exhibited significant differences among the evaluated irrigation regimes and potassium silicate treatments. RWC was reduced by increasing drought levels under moderate (75% ET) by 25.50% and severe (50% ET) drought by 31.54% compared to well-watered (100% ET) conditions. Otherwise, potassium silicate exogenous application enhanced RWC under moderate and severe drought stress conditions. The highest enhancement was assigned for the application of 20 g/L, which improved RWC by 50.0% under moderate drought and 38.1% under severe drought conditions compared to untreated control. The results of Gong et al. [26] and Kaya et al. [32] coincide with the obtained findings of this study, displaying that the exogenously applied potassium silicate enhanced RWC, indicating retention of water in cells that stimulated RWC under drought stress.

Squash plants exposed to moderate (75% ET) or severe (50% ET) drought stress displayed a substantial decline in the content of chlorophyll *a* by 2.6 and 4.5%, chlorophyll *b* by 2.1 and 4.1%, and carotenoids by 14.1 and 26.6%, compared well-watered conditions, in the same order (Figure 1). However, increasing levels of potassium silicate enhanced photosynthetic pigments under the two drought regimes compared with untreated control (control). The application of 20 g/L exhibited the highest improvement of chlorophyll *a* under moderate drought and severe drought by 5.8 and 4.9% compared with untreated control. Likewise, it enhanced chlorophyll *b* by 9.8 and 10.9% compared to untreated control under moderate and severe drought, respectively. Moreover, it boosted carotenoids by 54.5 and 28.6% compared to untreated control under moderate and severe drought, respectively. In this context, Rizwan et al. [30] disclosed that Si has a decisive role in mitigating drought stress through different mechanisms, such as enhancing phytohormone synthesis, uptake of mineral nutrients, osmotic adjustment, regulation of compatible solutes, modification of gas exchange attributes, and reduction in oxidative stress. Furthermore, Rubinowska et al. [49] and Shen et al. [50] manifested that Si is a beneficial element for various metabolic processes, as it suppresses chlorophyll degradation or enhances photosynthetic apparatus by promoting chlorophyll contents and water balance. Similarly, Pandey and Yadav [51] pointed out that spraying silicon increased chlorophyll content, water status, dry matter accumulation, dry matter production rate, and biological yield.

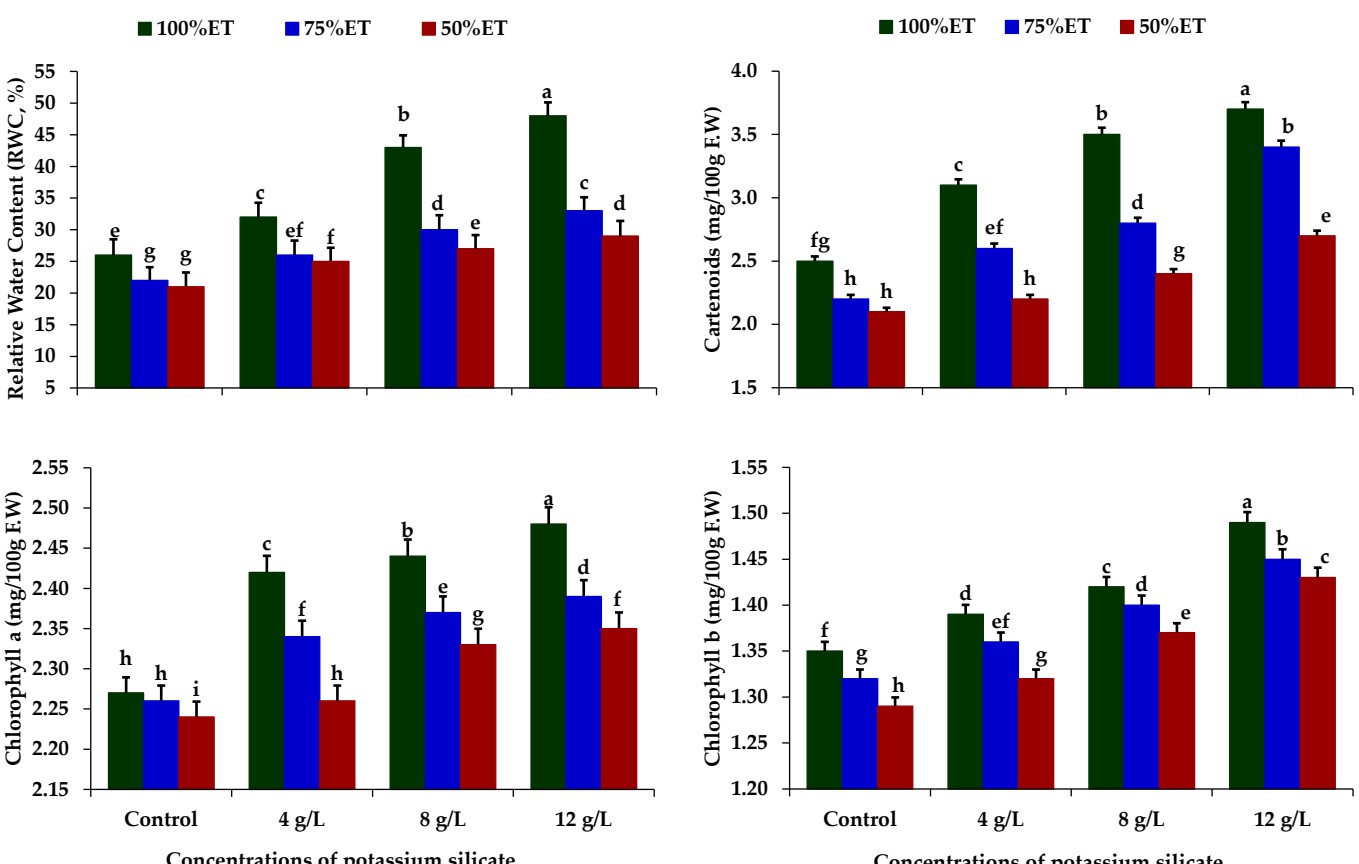

**Figure 1.** Impact of potassium silicate at different concentrations on RWC%, carotenoids, and chlorophyll content (*a*, *b*) of squash plants under different irrigation regimes. The bars on the columns represent SE and different letters are significantly different by Tukey HSD test at 0.05 significance level.

### 3.1.2. Antioxidant Enzyme Activities

Incremental changes in water stress levels from moderate (75% ET) to severe (50% ET) considerably enhanced the activity of peroxidase (POD), superoxidase dismutase (SOD), polyphenol oxidase (PPO), and catalase (CAT). Moderate drought substantially elevated POD, SOD, PPO, and CAT by 32.4, 14.1, 51.2, and 78.7% compared to well-watered conditions. While severe

drought increased POD, SOD, PPO, and CAT by 83.8, 60.0, 66.5, and 98.1%. Moreover, increasing the levels of potassium silicate from 10 to 20 g/L exhibited a significant increment of POD, PPO, SOD, and CAT (Table 1). The application of 20 g/L considerably stimulated POD, SOD, PPO, and CAT under moderate drought by 70.6, 70.5, 82.5, and 71.94% compared with untreated control, respectively. Likewise, under severe drought, it boosted POD, SOD, PPO, and CAT by 68.5, 88.7, 77.9, and 41.7% compared with untreated control, respectively. Various previous reports elucidated that the antioxidant defense system ameliorated drought tolerance by boosting the antioxidant enzyme activities [52]. In this context, Desoky et al. [53] demonstrated that the exogenously applied silicon promoted the activities of SOD, POD, and CAT under water scarcity conditions compared to untreated control. Furthermore, Ma [27] proved that silicon attenuated POD response, retained SOD adaptation, and elevated CAT activity under drought stress. Moreover, the enhancement of drought tolerance by silicon application is related to stimulating the antioxidant defense system. Accordingly, mitigating the oxidative damage induced by overproduction of ROS and maintaining various physiological processes under drought stress [25,54,55].

### 3.1.3. Mineral Content

Strengthening the drought level from moderate (75% ET) to severe (50% ET) displayed a substantial reduction in N%, P%, and K% of the shoot and root systems (Table 2). Moderate drought significantly declined N%, P%, and K% of the shoot by 12.6, 25.5, and 5.5% and of the root system by 18.7, 17.1, and 11.0%, compared to well-watered conditions, respectively. Severe drought notably reduced N%, P%, and K% of the shoot by 30.7, 50.9, and 23.6% and of the root system by 30.0, 19.5, and 19.5% compared to well-watered conditions, respectively. However, increasing potassium the silicate level from 10 to 20 g/L exhibited significant enhancement in the N, P, and K contents in plant root and shoot. The application of 20 g/L remarkably enhanced N, P, and K contents of the shoot by 32.5, 60.6, and 39.2% under moderate drought and by 25.8, 23.7, and 22.4% under severe drought compared to untreated control in the same order. Similarly, it boosted N, P, and K contents of the root under moderate drought by 30.3, 17.1, and 20.0%, and severe drought by 42.9, 26.5, and 14.5% compared to untreated control in the same order. In this context, Liang [56] elucidated that Si maintains the optimum supply of essential nutrients, which could be resulted by changing the soil pH. Moreover, Eneji et al. [57] indicated that silicon boosted the uptake of major essential elements in numerous types of grass that were evaluated under drought conditions. Besides, Yunus and Zari [58] elucidated that Si application caused a significant increase in $K^+$, $Ca^{2+}$, and $Mg^{2+}$ contents in tomato leaves.

### 3.2. Anatomical Characters

The moderate drought significantly decreased thickness of periderm, cortex thickness, midrib and mesophyll, vascular bundle of leaves (μ), collenchymatous tissue upper vascular bundle, and collenchymatous tissue lower vascular bundle thickness (μ), and number of xylem vessels per main vascular bundle of leaves by 25.9, 18.6, 11.5, 7.0, 21.7, 12.7, 24.4, and 22.8% compared to well-watered conditions (Table 3 and Figure 2). Moreover, severe drought significantly reduced the aforementioned parameters in the same order by 38.9, 45.3, 25.2, 15.8, 23.8, 22.5, 37.6, and 38.6% compared to well-watered conditions. However, potassium silicate in different levels (10 to 15 or 20 g/L) enhanced all studied anatomical measurements under moderate and severe drought compared to untreated control. In particular, the high level at 20 g/L considerably enhanced the thickness of periderm, cortex, midrib, mesophyll, vascular bundle, collenchymatous tissue upper and lower vascular bundle, and number of xylem vessels per main vascular bundle by 24.6, 37.0, 30.4, 41.1, 61.8, 41.7, 28.8, and 25.0% compared to untreated control. Similarly, it promoted the aforementioned parameters in the same order by 86.4, 39.2, 53.7, 49.0, 55.2, 88.9, 20.9, and 12.5% compared to untreated control. In this context, Shanan and El Sadek [59] manifested that the measurements of leaf thickness, upper epidermis thickness (μ), lower epidermis thickness (μ), and mesophyll thickness (μ) recorded remarkable reductions under drought stress. However, silicon application enhanced all these anatomical characteristics.

**Table 1.** Influence of potassium silicate on the activity of peroxidase (POD), superoxidase dismutase (SOD), polyphenol oxidase (PPO), and catalase (CAT) of squash plants under different irrigation regimes.

| Treatment | | 100% ET | 75% ET | 50% ET | Mean | 100% ET | 75% ET | 50% ET | Mean |
|---|---|---|---|---|---|---|---|---|---|
| | | Peroxidase ug/g (F.W) | | | | Superoxide Dismutase ug/g (F.W) | | | |
| Control | | 0.086 ± 0.006 [f] | 0.102 ± 0.01 [e] | 0.149 ± 0.008 [cd] | 0.112 ± 0.015 [C] | 0.109 ± 0.009 [f] | 0.163 ± 0.01 [e] | 0.213 ± 0.008 [de] | 0.162 ± 0.018 [D] |
| 10 g/L | | 0.104 ± 0.01 [e] | 0.133 ± 0.009 [de] | 0.164 ± 0.01 [c] | 0.134 ± 0.012 [B] | 0.167 ± 0.02 [e] | 0.168 ± 0.02 [e] | 0.239 ± 0.01 [cd] | 0.191 ± 0.012 [C] |
| 15 g/L | | 0.105 ± 0.007 [e] | 0.146 ± 0.006 [cd] | 0.207 ± 0.005 [b] | 0.153 ± 0.017 [B] | 0.201 ± 0.01 [de] | 0.234 ± 0.009 [cd] | 0.331 ± 0.07 [b] | 0.255 ± 0.014 [B] |
| 20 g/L | | 0.126 ± 0.004 [de] | 0.174 ± 0.01 [c] | 0.251 ± 0.004 [a] | 0.184 ± 0.015 [A] | 0.262 ± 0.07 [c] | 0.278 ± 0.02 [c] | 0.402 ± 0.06 [a] | 0.314 ± 0.016 [A] |
| Mean | | 0.105 ± 0.012 [B] | 0.139 ± 0.015 [B] | 0.193 ± 0.012 [A] | | 0.185 ± 0.018 [C] | 0.211 ± 0.015 [B] | 0.296 ± 0.02 [A] | |
| ANOVA | df | | | | *p*-value | | | | |
| Irrigation (I) | 2 | <0.001 | | | | <0.001 | | | |
| Treatment (T) | 3 | <0.001 | | | | <0.001 | | | |
| Season(S) | 1 | 0.168 | | | | 0.317 | | | |
| I × T | 6 | <0.001 | | | | <0.001 | | | |
| I × S | 2 | 0.976 | | | | 0.998 | | | |
| T × S | 3 | 0.988 | | | | 0.996 | | | |
| I × T × S | 6 | 0.869 | | | | 0.837 | | | |
| | | Polyphenol oxidase ug/g (F.W) | | | | Catalase ug/g (F.W) | | | |
| Control | | 0.136 ± 0.019 [f] | 0.730 ± 0.028 [e] | 0.871 ± 0.031 [d] | 0.579 ± 0.04 [C] | 0.065 ± 0.012 [f] | 0.417 ± 0.017 [d] | 0.521 ± 0.024 [c] | 0.334 ± 0.025 [D] |
| 10 g/L | | 0.878 ± 0.025 [d] | 1.27 ± 0.026 [b] | 1.09 ± 0.029 [c] | 1.079 ± 0.08 [B] | 0.348 ± 0.009 [e] | 0.510 ± 0.021 [c] | 0.621 ± 0.017 [b] | 0.493 ± 0.028 |
| 15 g/L | | 0.902 ± 0.024 [d] | 1.29 ± 0.022 [b] | 1.51 ± 0.036 [a] | 1.234 ± 0.04 [A] | 0.418 ± 0.015 [d] | 0.606 ± 0.02 [b] | 0.616 ± 0.022 [b] | 0.547 ± 0.022 [B] |
| 20 g/L | | 1.100 ± 0.026 [c] | 1.28 ± 0.027 [b] | 1.550 ± 0.031 [a] | 1.310 ± 0.03 [A] | 0.430 ± 0.020 [d] | 0.717 ± 0.006 [a] | 0.738 ± 0.009 [a] | 0.628 ± 0.021 [A] |
| Mean | | 0.754 ± 0.05 [C] | 1.14 ± 0.06 [B] | 1.255 ± 0.05 [A] | | 0.315 ± 0.023 [B] | 0.563 ± 0.026 [A] | 0.624 ± 0.015 [A] | |
| ANOVA | df | | | | *p*-value | | | | |
| Irrigation (I) | 2 | <0.001 | | | | <0.001 | | | |
| Treatment (T) | 3 | <0.001 | | | | <0.001 | | | |
| Season (S) | 1 | 0.342 | | | | 0.045 | | | |
| I × T | 6 | <0.001 | | | | <0.001 | | | |
| I × S | 2 | 0.337 | | | | 0.370 | | | |
| T × S | 3 | 0.366 | | | | 0.691 | | | |
| I × T × S | 6 | 0.730 | | | | 0.856 | | | |

[1] Different uppercase letters indicate significant difference among the main effect of irrigation regimes or potassium silicate concentrations at 0.05 significance level, while lowercase letter implies significant difference among their interaction.

**Table 2.** Influence of potassium silicate at different concentrations on N%, P%, and K% of squash shoot and root under different irrigation regimes.

| Treatment | | 100% ET | 75% ET | 50% ET | Mean | 100% ET | 75% ET | 50% ET | Mean |
|---|---|---|---|---|---|---|---|---|---|
| | | N % Shoot | | | | P% Shoot | | | |
| | Control | 3.61 ± 0.22 cd | 3.26 ± 0.22 e | 2.49 ± 0.20 g | 3.120 ± 0.41 D | 0.045 ± 0.003 c | 0.033 ± 0.008 e | 0.038 ± 0.004 d | 0.039 ± 0.012 C |
| | 10 g/L | 3.72 ± 0.18 c | 3.50 ± 0.20 d | 2.87 ± 0.21 f | 3.363 ± 0.30 C | 0.043 ± 0.004 c | 0.036 ± 0.005 de | 0.029 ± 0.008 f | 0.036 ± 0.013 D |
| | 15 g/L | 4.12 ± 0.20 b | 3.60 ± 0.23 cd | 2.80 ± 0.19 f | 3.507 ± 0.32 B | 0.055 ± 0.006 b | 0.043 ± 0.004 c | 0.024 ± 0.007 g | 0.041 ± 0.011 B |
| | 20 g/L | 5.10 ± 0.21 a | 4.10 ± 0.25 b | 3.30 ± 0.23 e | 4.167 ± 0.35 A | 0.075 ± 0.008 a | 0.053 ± 0.007 b | 0.029 ± 0.005 f | 0.052 ± 0.009 A |
| | Mean | 4.14 ± 0.34 A | 3.62 ± 0.38 B | 2.87 ± 0.31 C | | 0.055 ± 0.004 A | 0.041 ± 0.007 B | 0.027 ± 0.009 C | |
| ANOVA | df | | | | *p*-value | | | | |
| Irrigation (I) | 2 | <0.001 | | | | <0.001 | | | |
| Treatment (T) | 3 | <0.001 | | | | <0.001 | | | |
| Season (S) | 1 | 0.072 | | | | 0.183 | | | |
| I × T | 6 | <0.001 | | | | <0.001 | | | |
| I × S | 2 | 0.915 | | | | 0.538 | | | |
| T × S | 3 | 0.636 | | | | 0.994 | | | |
| I × T × S | 6 | 0.570 | | | | 0.952 | | | |
| | | K% shoot | | | | N % root | | | |
| | Control | 0.86 ± 0.11 d | 0.79 ± 0.22 e | 0.76 ± 0.22 ef | 0.810 ± 0.21 C | 1.88 ± 0.24 f | 1.65 ± 0.22 h | 1.35 ± 0.21 j | 1.63 ± 0.33 D |
| | 10 g/L | 0.96 ± 0.21 c | 0.93 ± 0.23 cd | 0.72 ± 0.20 f | 0.870 ± 0.21 B | 1.93 ± 0.23 d | 1.73 ± 0.19 g | 1.50 ± 0.25 i | 1.72 ± 0.34 C |
| | 15 g/L | 1.29 ± 0.20 a | 1.10 ± 0.25 b | 0.93 ± 0.23 cd | 1.107 ± 0.25 A | 2.83 ± 0.21 a | 1.94 ± 0.26 e | 1.64 ± 0.24 h | 2.14 ± 0.35 B |
| | 20 g/L | 1.29 ± 0.23 a | 1.10 ± 0.28 b | 0.93 ± 0.24 cd | 1.107 ± 0.26 A | 2.54 ± 0.22 b | 2.15 ± 0.22 c | 1.93 ± 0.23 e | 2.21 ± 0.32 A |
| | Mean | 1.10 ± 0.29 A | 1.04 ± 0.32 B | 0.84 ± 0.28 C | | 2.30 ± 0.29 A | 1.87 ± 0.31 B | 1.61 ± 0.35 C | |
| ANOVA | df | | | | *p*-value | | | | |
| Irrigation (I) | 2 | <0.001 | | | | <0.001 | | | |
| Treatment (T) | 3 | <0.001 | | | | <0.001 | | | |
| Season (S) | 1 | 0.771 | | | | 0.061 | | | |
| I × T | 6 | <0.001 | | | | <0.001 | | | |
| I × S | 2 | 0.825 | | | | 0.952 | | | |
| T × S | 3 | 0.886 | | | | 0.406 | | | |
| I × T × S | 6 | 0.706 | | | | 0.058 | | | |
| | | P % root | | | | K% root | | | |
| | Control | 0.035 ± 0.008 d | 0.035 ± 0.003 d | 0.032 ± 0.003 e | 0.034 ± 0.005 C | 0.75 ± 0.22 e | 0.65 ± 0.21 g | 0.62 ± 0.22 h | 0.67 ± 0.41 D |
| | 10 g/L | 0.054 ± 0.004 b | 0.026 ± 0.001 f | 0.013 ± 0.005 h | 0.031 ± 0.007 D | 0.79 ± 0.23 c | 0.76 ± 0.23 de | 0.62 ± 0.20 h | 0.72 ± 0.42 C |
| | 15 g/L | 0.053 ± 0.003 b | 0.031 ± 0.005 e | 0.022 ± 0.002 g | 0.035 ± 0.008 B | 0.84 ± 0.21 b | 0.74 ± 0.25 e | 0.67 ± 0.19 g | 0.75 ± 0.38 B |
| | 20 g/L | 0.024 ± 0.007 fg | 0.041 ± 0.004 c | 0.065 ± 0.001 a | 0.043 ± 0.006 A | 0.90 ± 0.24 a | 0.78 ± 0.23 cd | 0.71 ± 0.24 f | 0.80 ± 0.35 A |
| | Mean | 0.041 ± 0.008 A | 0.034 ± 0.01 B | 0.033 ± 0.007 C | | 0.82 ± 0.29 A | 0.73 ± 0.32 B | 0.66 ± 0.35 C | |
| ANOVA | df | | | | *p*-value | | | | |
| Irrigation (I) | 2 | <0.001 | | | | <0.001 | | | |
| Treatment (T) | 3 | <0.001 | | | | <0.001 | | | |
| Season (S) | 1 | 0.059 | | | | 0.679 | | | |
| I × T | 6 | <0.001 | | | | <0.001 | | | |
| I × S | 2 | 0.046 | | | | 0.402 | | | |
| T × S | 3 | 0.052 | | | | 0.267 | | | |
| I × T × S | 6 | 0.093 | | | | 0.608 | | | |

[1] Different uppercase letters indicate significant difference among the main effect of irrigation regimes or potassium silicate concentrations at 0.05 significance level, while lowercase letter implies significant difference among their interaction.

**Table 3.** Influence of potassium silicate at different concentrations on anatomical characters of squash plants under different irrigation regimes.

| Treatment | | | 100% ET | 75% ET | 50% ET | Mean | 100% ET | 75% ET | 50% ET | Mean |
|---|---|---|---|---|---|---|---|---|---|---|
| | | | Thickness of Periderm (μ) | | | | Thickness of Cortex (μ) | | | |
| | Control | | 102 ± 2.11 b | 69.0 ± 3.04 f | 44.0 ± 3.21 g | 71.7 ± 4.21 D | 202 ± 3.23 d | 162 ± 4.30 e | 102 ± 3.71 h | 155.3 ± 5.72 D |
| | 10 g/L | | 103 ± 3.93 b | 83.0 ± 1.96 d | 69.0 ± 2.62 f | 85.0 ± 3.42 B | 203 ± 5.29 d | 162 ± 4.92 e | 117 ± 3.89 g | 160.0 ± 4.21 C |
| | 15 g/L | | 102 ± 2.61 b | 82.0 ± 3.31 d | 72.0 ± 2.78 e | 85.3 ± 3.91 B | 252 ± 4.64 b | 202 ± 5.78 d | 142 ± 4.53 f | 198.7 ± 6.18 B |
| | 20 g/L | | 126 ± 2.81 a | 86.0 ± 2.61 c | 82.0 ± 3.69 d | 98.0 ± 3.89 A | 262 ± 6.17 a | 222 ± 5.89 c | 142 ± 4.67 f | 208.7 ± 5.33 A |
| | Mean | | 108.0 ± 4.23 A | 80.0 ± 4.57 B | 66.08 ± 4.75 C | | 229.8 ± 7.23 A | 187.0 ± 6.23 B | 125.8 ± 6.23 C | |
| ANOVA | | df | | | | *p*-value | | | | |
| Irrigation (I) | | 2 | <0.001 | | | | <0.001 | | | |
| Treatment (T) | | 3 | <0.001 | | | | <0.001 | | | |
| I × T | | 6 | <0.001 | | | | <0.001 | | | |
| | | | Thickness of midrib (μ) | | | | Thickness of mesophyll (μ) | | | |
| | Control | | 542 ± 4.20 j | 572 ± 5.61 h | 402 ± 4.32 l | 557.0 ± 6.70 D | 123 ± 2.13 ef | 112 ± 2.20 g | 102 ± 2.29 h | 112.3 ± 4.21 D |
| | 10 g/L | | 702 ± 3.18 d | 591 ± 4.23 g | 412 ± 3.98 k | 568.3 ± 4.12 C | 123 ± 2.16 ef | 122 ± 1.90 f | 102 ± 2.30 h | 115.7 ± 3.20 C |
| | 15 g/L | | 767 ± 4.69 b | 609 ± 5.37 f | 566 ± 5.13 I | 647.0 ± 6.56 B | 136 ± 3.15 d | 125 ± 2.16 e | 112 ± 3.17 g | 124.3 ± 3.16 B |
| | 20 g/L | | 833 ± 5.18 a | 746 ± 4.08 c | 618 ± 3.37 e | 732.0 ± 5.92 A | 174 ± 2.98 a | 158 ± 3.11 b | 152 ± 2.41 c | 161.3 ± 3.81 A |
| | Mean | | 711 ± 7.86 A | 629.5 ± 6.40 B | 532.0 ± 5.82 C | | 139.0 ± 4.35 A | 129.3 ± 3.38 B | 117.0 ± 3.65 C | |
| ANOVA | | df | | | | *p*-value | | | | |
| Irrigation (I) | | 2 | <0.001 | | | | <0.001 | | | |
| Treatment (T) | | 3 | <0.001 | | | | <0.001 | | | |
| I × T | | 6 | <0.001 | | | | <0.001 | | | |
| | | | Thickness of vascular bundle (μ) | | | | No. of xylem vessel/main | | | |
| | Control | | 202 ± 1.16 g | 152 ± 1.18 j | 134 ± 1.41 k | 162.7 ± 2.31 D | 23.0 ± 1.72 de | 20.0 ± 1.23 f | 16.0 ± 0.56 g | 19.7 ± 2.78 D |
| | 10 g/L | | 223 ± 2.62 d | 167 ± 1.52 i | 166 ± 1.50 i | 185.3 ± 2.55 C | 27.0 ± 1.63 c | 23.0 ± 1.48 de | 18.0 ± 0.89 fg | 22.7 ± 2.25 C |
| | 15 g/L | | 252 ± 1.17 b | 218 ± 1.13 e | 191 ± 1.67 h | 220.3 ± 2.01 B | 29.0 ± 1.08 b | 22.0 ± 1.69 e | 18.0 ± 1.01 fg | 23.0 ± 1.68 B |
| | 20 g/L | | 282 ± 1.98 a | 246 ± 1.19 c | 208 ± 1.29 f | 245.0 ± 2.09 A | 35.0 ± 1.16 a | 25.0 ± 1.56 d | 18.0 ± 1.03 fg | 26.0 ± 2.34 A |
| | Mean | | 240 ± 3.56 A | 188 ± 2.19 B | 183 ± 2.25 C | | 28.5 ± 2.40 A | 22.0 ± 1.89 B | 17.5 ± 2.95 C | |
| ANOVA | | df | | | | *p*-value | | | | |
| Irrigation (I) | | 2 | <0.001 | | | | <0.001 | | | |
| Treatment (T) | | 3 | <0.001 | | | | <0.001 | | | |
| I × T | | 6 | <0.001 | | | | <0.001 | | | |
| | | | Thickness of collenchymatous tissue and upper epidermis (μ) | | | | Thickness of collenchymatous tissue and lower epidermis (μ) | | | |
| | Control | | 92.0 ± 0.72 d | 72.0 ± 0.78 f | 54.0 ± 0.66 g | 72.7 ± 1.71 D | 64.0 ± 0.62 b | 52.0 ± 0.65 g | 43.0 ± 0.78 h | 53.0 ± 1.71 B |
| | 10 g/L | | 92.0 ± 0.59 d | 83.0 ± 0.89 e | 71.0 ± 0.75 f | 82.0 ± 1.16 C | 72.0 ± 0.68 e | 58.0 ± 0.53 f | 45.0 ± 0.73 g | 58.3 ± 1.54 B |
| | 15 g/L | | 100 ± 0.65 b | 81.0 ± 0.89 e | 73.0 ± 0.79 f | 84.7 ± 1.28 B | 78.0 ± 0.51 c | 54.0 ± 0.48 g | 51.0 ± 0.66 g | 61.0 ± 1.17 B |
| | 20 g/L | | 103 ± 0.69 a | 102 ± 0.75 c | 102 ± 0.89 c | 102.3 ± 2.06 A | 92.0 ± 0.73 a | 67.0 ± 0.76 d | 52.0 ± 0.70 h | 70.3 ± 1.43 A |
| | Mean | | 96.8 ± 1.72 A | 84.5 ± 1.82 B | 75.0 ± 1.78 C | | 76.5 ± 1.72 A | 57.8 ± 1.56 C | 47.8 ± 1.74 B | |
| ANOVA | | df | | | | *p*-value | | | | |
| Irrigation (I) | | 2 | <0.001 | | | | <0.001 | | | |
| Treatment (T) | | 3 | <0.001 | | | | <0.001 | | | |
| I × T | | 6 | <0.001 | | | | <0.001 | | | |

[1] Different uppercase letters indicate significant difference among the main effect of irrigation regimes or potassium silicate concentrations at 0.05 significance level, while lowercase letter implies significant difference among their interaction.

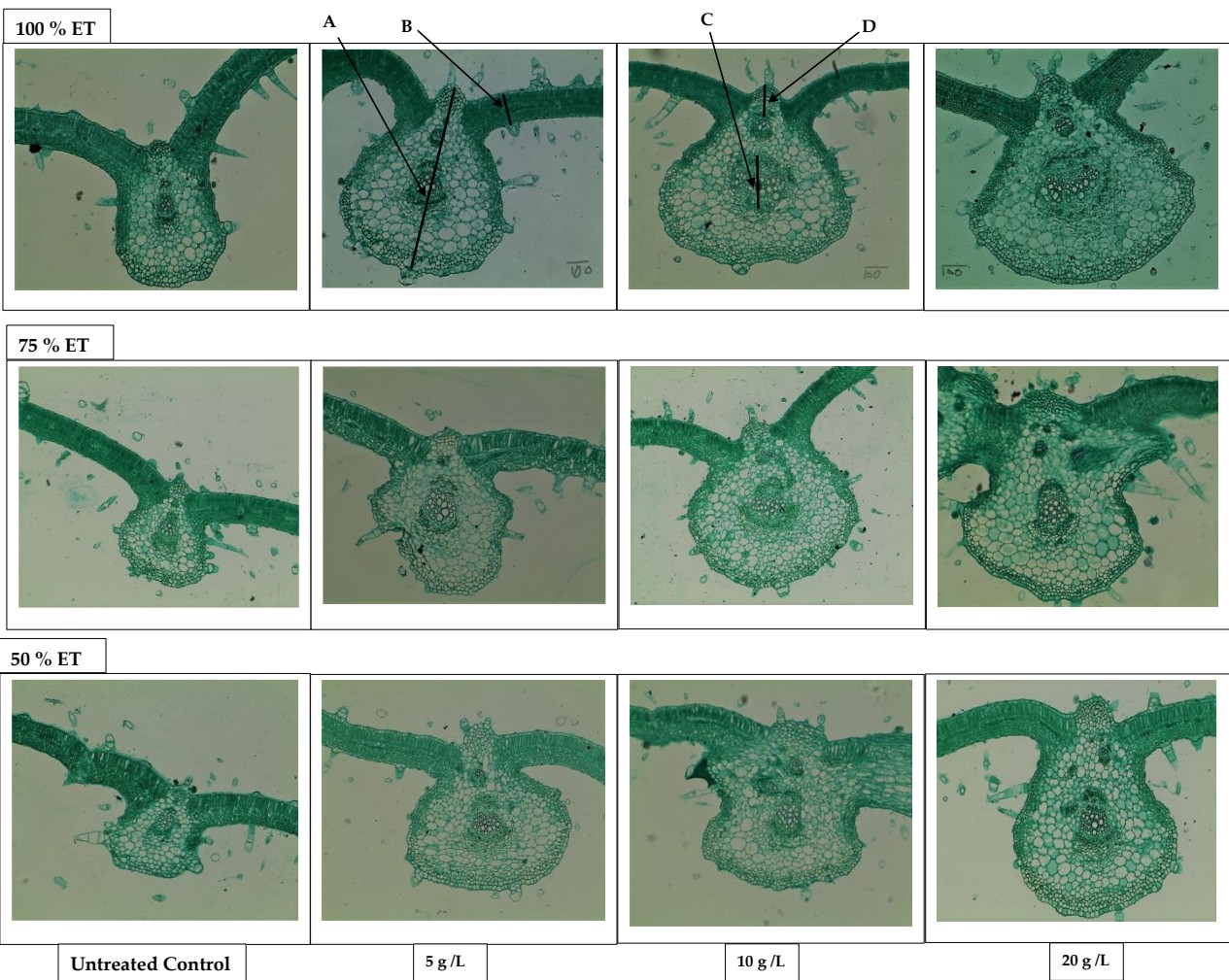

**Figure 2.** Cross-sections of squash leaves under potassium silicate concentrations and irrigation regimes. (A: Midrib thickness (μ), B: Mesophyll thickness (μ), C: Vascular bundle thickness (μ), and D: Thickness of collenchymatous tissue and epidermis (μ). Bar= 100μ).

*3.3. Growth Parameters*

Moderate (75% ET) and severe (50% ET) drought stress significantly declined all growth and yield traits, i.e., plant height, fresh and dry weight of root and shoot, number of leaves per plant, and fruit yield (Table 4). Moderate drought significantly declined plant height, fresh and dry weight of shoot, number of leaves per plant, and fruit yield by 17.6, 14.9, 1.6, 2.9, and 20.4%, compared to well-watered conditions, respectively. Similarly, severe drought considerably diminished plant height, number of leaves per plant, and fruit yield by 32.5, 22.6, 4.1, 11.8, and 31.0% compared to well-watered conditions, respectively. Otherwise, increasing potassium silicate level from 10 g/L to 15 g/L or 20 g/L considerably boosted all growth and yield traits compared with untreated control (Table 4). The growth and yield traits of the plants treated with 20 g/L potassium silicate under moderate or severe drought stress surpassed the treated plants with 15 or 10 g/L. The application of 20 g/L potassium silicate reinforced plant height, number of leaves per plant, and fruit yield by 61.6, 70.5, 9.6, 46.2, and 49.0% compared to untreated plants. Likewise, under severe drought, it enhanced plant height, number of leaves per plant, and fruit yield by 76.0, 88.3, 9.0, 50.0, and 83.9% compared to untreated plants. Previous researchers documented the benefits of foliar-applied Si under drought stress. In this context, Gong et al. [26] disclosed a considerable enhancement in plant growth of treated wheat by Si under well-watered and drought stress conditions. Likewise, Crusciol et al. [60] proved that exogenously sprayed Si boosted potato production, due to enhancement of the accumulation of total sugars and proline under drought stress conditions.

**Table 4.** Influence of potassium silicate on plant height, number of leaves per plant, fresh and dry weights of shoot/plant of squash plants under different irrigation regimes.

| Treatment | df | 100% ET | 75% ET | 50% ET | Mean | 100% ET | 75% ET | 50% ET | Mean |
|---|---|---|---|---|---|---|---|---|---|
| | | Root Length (cm) | | | | Root Fresh Weight per Plant (g) | | | |
| Control | | 22.0 ± 1.12 [d] | 23.0 ± 1.65 [d] | 25.0 ± 2.20 [cd] | 23.3 ± 2.66 [D] | 3.0 ± 0.89 [c] | 6.0 ± 0.88 [b] | 3.0 ± 0.66 [c] | 4.0 ± 1.21 [B] |
| 10 g/L | | 21.0 ± 1.18 [d] | 26.0 ± 1.35 [cd] | 34.0 ± 1.54 [bc] | 27.0 ± 2.36 [C] | 6.0 ± 0.78 [b] | 5.0 ± 0.79 [bc] | 6.0 ± 0.73 [b] | 5.7 ± 1.37 [A] |
| 15 g/L | | 33.0 ± 1.52 [bc] | 35.0 ± 1.26 [b] | 30.0 ± 1.60 [b–d] | 32.7 ± 1.96 [B] | 6.0 ± 0.65 [b] | 7.0 ± 1.01 [ab] | 8.0 ± 0.82 [ab] | 7.0 ± 1.18 [A] |
| 20 g/L | | 30.1 ± 1.36 [b–d] | 30.0 ± 1.87 [b–d] | 49.0 ± 1.97 [a] | 36.3 ± 2.73 [A] | 6.0 ± 0.77 [b] | 7.0 ± 0.42 [ab] | 9.0 ± 0.74 [a] | 7.3 ± 1.64 [A] |
| Mean | | 26.5 ± 2.45 [C] | 28.5 ± 2.53 [B] | 34.5 ± 2.83 [A] | | 5.3 ± 1.27 [b] | 6.3 ± 1.08 [ab] | 6.5 ± 1.64 [a] | |
| ANOVA | df | | | | *p*-value | | | | |
| Irrigation (I) | 2 | <0.001 | | | | <0.001 | | | |
| Treatment (T) | 3 | <0.001 | | | | <0.001 | | | |
| Season (S) | 1 | 0.381 | | | | 0.831 | | | |
| I × T | 6 | <0.001 | | | | <0.001 | | | |
| I × S | 2 | 0.047 | | | | 0.049 | | | |
| T × S | 3 | 0.036 | | | | 0.394 | | | |
| I × T × S | 6 | 0.052 | | | | 0.106 | | | |
| | | Root dry weight per plant (g) | | | | Plant height (cm) | | | |
| Control | | 1.3 ± 0.21 [b] | 1.7 ± 0.16 [b] | 1.3 ± 0.12 [b] | 1.4 ± 0.26 [B] | 45.0 ± 2.26 [bc] | 32.0 ± 2.18 [de] | 25.0 ± 2.74 [e] | 34.0 ± 3.72 [C] |
| 10 g/L | | 1.3 ± 0.18 [b] | 2.0 ± 0.19 [a] | 2.3 ± 0.15 [ab] | 2.3 ± 0.22 [A] | 53.0 ± 2.58 [ab] | 40.0 ± 3.29 [cd] | 34.0 ± 2.19 [cd] | 42.3 ± 3.69 [B] |
| 15 g/L | | 2.0 ± 0.12 [ab] | 2.0 ± 0.22 [ab] | 3.0 ± 0.26 [ab] | 2.3 ± 25 [A] | 56.0 ± 3.18 [a] | 52.0 ± 3.15 [ab] | 42.0 ± 2.64 [b–d] | 50.0 ± 3.99 [A] |
| 20 g/L | | 2.0 ± 0.14 [ab] | 3.3 ± 0.31 [a] | 3.0 ± 0.23 [ab] | 2.3 ± 0.23 [A] | 59.0 ± 3.20 [a] | 51.7 ± 3.27 [ab] | 44.0 ± 3.18 [bc] | 51.6 ± 3.22 [A] |
| Mean | | 1.7.0 ± 0.29 [B] | 2.3 ± 0.35 [A] | 2.4 ± 0.29 [A] | | 53.3 ± 4.30 [A] | 43.9 ± 4.97 [B] | 36.0 ± 4.69 [C] | |
| ANOVA | df | | | | *p*-value | | | | |
| Irrigation (I) | 2 | <0.001 | | | | <0.001 | | | |
| Treatment (T) | 3 | <0.001 | | | | <0.001 | | | |
| Season (S) | 1 | 0.278 | | | | 0.402 | | | |
| I × T | 6 | <0.001 | | | | <0.001 | | | |
| I × S | 2 | 0.481 | | | | 0.529 | | | |
| T × S | 3 | 0.308 | | | | 0.039 | | | |
| I × T × S | 6 | 0.093 | | | | 0.063 | | | |

**Table 4.** *Cont.*

| Treatment | df | 100% ET | 75% ET | 50% ET | Mean | 100% ET | 75% ET | 50% ET | Mean |
|---|---|---|---|---|---|---|---|---|---|
| | | Shoot fresh weight per plant (g) | | | | Shoot dry weight per plant (g) | | | |
| Control | | 392 ± 3.28 [j] | 356 ± 6.18 [k] | 315 ± 6.72 [l] | 354.3 ± 5.39 [D] | 116 ± 1.23 [e] | 114 ± 1.62 [f] | 111 ± 1.97 [g] | 113.7 ± 1.61 [C] |
| 10 g/L | | 556 ± 4.90 [f] | 491 ± 5.27 [h] | 432 ± 5.19 [i] | 493.0 ± 5.12 [C] | 122 ± 1.61 [c] | 119 ± 2.18 [d] | 116 ± 1.76 [e] | 119.0 ± 1.85 [B] |
| 15 g/L | | 666 ± 5.16 [b] | 575 ± 5.96 [e] | 505 ± 5.32 [g] | 582.0 ± 5.48 [B] | 128 ± 1.23 [a] | 125 ± 1.78 [b] | 120 ± 1.41 [cd] | 124.3 ± 1.47 [A] |
| 20 g/L | | 771 ± 5.28 [a] | 607 ± 7.26 [c] | 593 ± 6.48 [d] | 657.0 ± 6.27 [A] | 125 ± 1.45 [b] | 125 ± 2.67 [b] | 121 ± 1.84 [cd] | 123.7 ± 1.99 [A] |
| Mean | | 596.3 ± 4.66 [A] | 507.3 ± 6.17 [B] | 461.3 ± 5.74 [C] | | 122.8 ± 1.38 [A] | 120.8 ± 2.06 [B] | 117 ± 1.75 [C] | |
| ANOVA | df | | | | *p*-value | | | | |
| Irrigation (I) | 2 | <0.001 | | | | <0.001 | | | |
| Treatment (T) | 3 | <0.001 | | | | <0.001 | | | |
| Season (S) | 1 | 0.228 | | | | 0.295 | | | |
| I × T | 6 | <0.001 | | | | <0.001 | | | |
| I × S | 2 | 0.048 | | | | 0.027 | | | |
| T × S | 3 | 0.442 | | | | 0.867 | | | |
| I × T × S | 6 | 0.029 | | | | 0.037 | | | |
| | | Number of leaves per plant | | | | Fruit weight per plant (g) | | | |
| Control | | 15.0 ± 0.73 [bc] | 13.0 ± 0.77 [c] | 12.0 ± 0.89 [c] | 13.3 ± 1.80 [C] | 865 ± 3.21 [f] | 633 ± 2.17 [j] | 502 ± 2.78 [l] | 666.7 ± 3.72 [D] |
| 10 g/L | | 17.0 ± 0.67 [b] | 14.0 ± 0.64 [bc] | 13.0 ± 0.75 [c] | 14.7 ± 1.69 [B] | 954 ± 2.18 [c] | 797 ± 2.23 [h] | 625 ± 2.73 [k] | 792.0 ± 3.68 [C] |
| 15 g/L | | 17.0 ± 1.01 [b] | 20.0 ± 0.23 [a] | 17.0 ± 0.67 [b] | 18.7 ± 1.62 [A] | 1124 ± 2.13 [a] | 824 ± 2.19 [g] | 724 ± 2.31 [i] | 890.7 ± 3.39 [B] |
| 20 g/L | | 19.0 ± 0.93 [b] | 19.0 ± 0.78 [b] | 18.0 ± 0.93 [b] | 18.7 ± 1.44 [A] | 1074 ± 3.14 [b] | 943 ± 3.14 [d] | 923 ± 2.56 [e] | 980.0 ± 3.63 [A] |
| Mean | | 17.0 ± 1.54 [A] | 16.5 ± 1.21 [A] | 15.0 ± 1.61 [B] | | 1004 ± 3.79 [A] | 799.3 ± 3.43 [B] | 693.5 ± 3.60 [B] | |
| ANOVA | df | *p*-value | | | | | | | |
| Irrigation (I) | 2 | <0.001 | | | | <0.001 | | | |
| Treatment (T) | 3 | <0.001 | | | | <0.001 | | | |
| Season (S) | 1 | 0.153 | | | | 0.052 | | | |
| I × T | 6 | <0.001 | | | | <0.001 | | | |
| I × S | 2 | 0.084 | | | | 0.264 | | | |
| T × S | 3 | 0.047 | | | | 0.296 | | | |
| I × T × S | 6 | 0.059 | | | | 0.439 | | | |

[1] Different uppercase letters indicate significant difference among the main effect of irrigation regimes or potassium silicate concentrations at 0.05 significance level, while lowercase letter implies significant difference among their interaction.

## 4. Conclusions

Exposing squash plants to moderate (75% ET) or severe (50% ET) drought stress gradually decreased all evaluated physiological parameters, anatomical characteristics, and growth traits compared with well-watered (100% ET) conditions. However, exogenously sprayed potassium silicate mitigated the devastating impacts on all evaluated traits. In particular, the application at 20 g/L was highly effective in enhancing all physiological parameters, anatomical characteristics, and growth traits under moderate and severe drought conditions. Conclusively, the exogenous foliar application of potassium silicate at 20 g/L could be more effective in promoting drought tolerance, which can be employed in reducing the losses caused by drought stress in squash growing regions.

**Supplementary Materials:** The following supporting information can be downloaded at: https://www.mdpi.com/article/10.3390/agronomy12092155/s1, Table S1: Monthly average temperature, relative humidity, dew point, precipitation and wind speed in 2019 and 2020 growing seasons.

**Author Contributions:** Conceptualization, E.S.A. and A.M.E.-T.; methodology, E.S.A. and A.M.E.-T.; software, K.S.A., M.A.M.A. and M.M.A.; validation, E.S.A., K.S.A., M.A.M.A., M.M.A. and A.M.E.-T.; formal analysis, K.S.A., M.A.M.A. and M.M.A.; investigation, E.S.A. and A.M.E.-T.; resources, K.S.A., M.A.M.A., F.A.S., S.M.A., T.A.A.E.-M. and M.M.A.; data curation, E.S.A., K.S.A., M.A.M.A., M.M.A. and A.M.E.-T.; writing—original draft preparation, E.S.A., F.A.S., S.M.A. and T.A.A.E.-M.; writing—review and editing, K.S.A., M.A.M.A., F.A.S., S.M.A., T.A.A.E.-M. and M.M.A.; funding acquisition, K.S.A., M.A.M.A., F.A.S., S.M.A., T.A.A.E.-M. and M.M.A. All authors have read and agreed to the published version of the manuscript.

**Funding:** This research was funded by Princess Nourah bint Abdulrahman University Researchers Supporting Project number (PNURSP2022R318), Princess Nourah bint Abdulrahman University, Riyadh, Saudi Arabia.

**Institutional Review Board Statement:** Not applicable.

**Informed Consent Statement:** Not applicable.

**Data Availability Statement:** The data presented in this study are available upon request from the corresponding author.

**Acknowledgments:** The authors would like to thank Princess Nourah bint Abdulrahman University Researchers Supporting Project number (PNURSP2022R318), Princess Nourah bint Abdulrahman University, Riyadh, Saudi Arabia. The co-author Khalid S. Alshallash from Saudi Arabia would like to thank the deanship of scientific research at Imam Mohammed Bin Saud Islamic University for supporting publication of this research work. Also, the co-author Mesfer M. Alqahtani would like to thank the deanship of scientific research at Shaqra University for supporting this research work.

**Conflicts of Interest:** The authors declare no conflict of interest.

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
