# Peer review of "Physiological, Anatomical, and Agronomic Responses of Cucurbita pepo to Exogenously Sprayed Potassium Silicate at Different Concentrations under Varying Water Regimes"

_agronomy, doi:10.3390/agronomy12092155_

Round 1

Reviewer 1 Report

The present article assessed the impact of potassium silicate on physiological, anatomical, and agronomic traits of cucurbita pepo in response to the limited water supply. The overall presentation is good with significant findings. After substantial changes, the article can be acceptable for publication in Agronomy Journal.

 I have some comments and suggestions for authors to address

 Ø  Line 25-26:  Authors write as “gradually declined photosynthetic pigments, relative water content (RWC), mineral content (N, P, and K) physiological parameters…………” Plz mention % increase or decrease value for clarity

Ø  Line 24: Plz correct as 100, 75 and 50%.........

Ø  Line 29: Plz mention % increase value against control “…… the antioxidants enzymes activities POD, PPO, SOD, and CAT significantly increased……….”

Ø  Line 20-38: Abstract needs improvement

Ø  Line 42-75: The introduction section is fine but needs minor

Ø  Line 92: Please mention climatic data during the experiment (Spring season 2019 and 2020)

Ø  Line 91-95: When Si sprayed means how many days interval after drought stress starts?

Ø  Line 93: Authors choose 10, 15, and 20 g/L K2SiO3 concentration, not Si, right?

Ø  Line 93-95: Sentence meaning not clear, plz revise

Ø  Line 104: Measured parameters??? Plz delete

Ø  Line 107: Authors choose 3rd leaf for photosynthetic pigment analysis, why not 2 or 4 leaf? Any specific reason. How to know 3rd leaf is photosynthetically mature for analysis?

Ø  Line 110: Why not choose the same leaf (3rd) for enzyme analysis?

Ø  Line 106-118: The methods are not written or carried out correctly, so they need to be better explained and corrected. Particularly the enzyme activities need to be explained and corrected

Ø  Line 144: Plz correct the sentence error

Ø  Results and Discussion section needs critical revision. Only authors mention enhanced or reduced, or decreased, not a single place mentions the % increase or decrease value for clarity in comparison to control

Ø  Line 288: Plz incorporate the significance of the study

Author Response

Dear Editor,

We would like to thank you and the reviewers for the time and efforts devoted to our manuscript entitled “Physiological, Anatomical, and Agronomic Responses of Cucurbita pepo to Exogenously Sprayed Potassium Silicate at Different Concentrations under Varying Water Regimes” (agronomy-1856718). We have revised the manuscript according to the comments and suggestions pointed out by the reviewers. We have addressed the comments of the reviewers in point-by-point below in red color; in addition, we have highlighted all the associated changes made to the manuscript using track changes.

Yours sincerely,

Authors

Reviewer 1

The present article assessed the impact of potassium silicate on physiological, anatomical, and agronomic traits of cucurbita pepo in response to the limited water supply. The overall presentation is good with significant findings. After substantial changes, the article can be acceptable for publication in Agronomy Journal.

Re: We would like to thank Reviewer 1 for his/her time dedicated to our manuscript. We highly appreciate his/her positive assessment of our work. We appreciate his/her comment to improve the quality of the manuscript.

I have some comments and suggestions for authors to address

 Ø  Line 25-26:  Authors write as “gradually declined photosynthetic pigments, relative water content (RWC), mineral content (N, P, and K) physiological parameters…………” Plz mention % increase or decrease value for clarity

Re: The percentages of increase or decrease have been added for all evaluated parameters in the result section, please see lines 191-210, 229-237, 257-268, and 290-312 in the revised version but it is difficult to be added to the abstract.

Ø  Line 24: Plz correct as 100, 75 and 50%.........

Re: Done as requested ( line 24)

Ø  Line 29: Plz mention % increase value against control “…… the antioxidants enzymes activities POD, PPO, SOD, and CAT significantly increased……….”

Re: The percentages of increase or decrease have been added in the result section, but it is difficult to be added to the abstract.

Ø  Line 20-38: Abstract needs improvement

Re: The abstract has been revised and improved as suggested

Ø  Line 42-75: The introduction section is fine but needs minor

Re: The introduction has been revised and improved as suggested

Ø  Line 92: Please mention climatic data during the experiment (Spring season 2019 and 2020)

Re: The metrological data has been added as supplementary materials (Table S1)

Ø  Line 91-95: When Si sprayed means how many days interval after drought stress starts?

Re: More details have been added (lines 103-104)

Ø  Line 93: Authors choose 10, 15, and 20 g/L K2SiO3 concentration, not Si, right?

Re: The used substance was potassium silicate K2SiO3 42% as a source of Si

Ø  Line 93-95: Sentence meaning not clear, plz revise

Re: Done as suggested please (lines 102-111)

Ø  Line 104: Measured parameters??? Plz delete

Re: Done as suggested (113)

Ø  Line 107: Authors choose 3rd leaf for photosynthetic pigment analysis, why not 2 or 4 leaf? Any specific reason. How to know 3rd leaf is photosynthetically mature for analysis?

Re: The main purpose was to unify the place of taking the sample.

Ø  Line 110: Why not choose the same leaf (3rd) for enzyme analysis?

Re: The reason for selecting the second leaf for enzyme analysis was to avoid the incomplete growth of vascular tissue in the third leaf.

Ø  Line 106-118: The methods are not written or carried out correctly, so they need to be better explained and corrected. Particularly the enzyme activities need to be explained and corrected

Re: The methods have been revised and more details have been added (lines 115-150)

Ø  Results and Discussion section needs critical revision. Only authors mention enhanced or reduced, or decreased, not a single place mentions the % increase or decrease value for clarity in comparison to control

Re: The percentages of increase or decrease for all evaluated parameters have been added in the result section, please see lines 191-210, 229-237, 257-268, and 290-312 in the revised version.

Ø  Line 288: Plz incorporate the significance of the study.

Re: Done as suggested (line 369-371)

Reviewer 2 Report

Dear Authors,

Overall, I enjoyed reading this well written manuscript. The results are clearly presented and well discussed. I recommend acceptance of this work after minor revision or addressing a few comments below:

Lines 88 to 90. Would be great to show how much volume of water per hour being employed in Drip Irrigation.

Line 99. Please provide the key standard agronomic practices of the region. “Standard agronomic practices (i.e. xxxxxxxxxxxxx?) were performed”

Lines 110-111 Please provide reference to the process being used in the study.

Lines 130-140 Better to provide references for the described processes if possible.

 Results:

The Tables and Figures showing the results/data were appropriately presented. I noticed that the results showed significant interactions (Irrigation X Si treatments) in most parameters (Physiological, Anatomical and Agronomic) used in the study. However, this interaction between Irrigation and Si treatments was barely mentioned or emphasized in the results/discussions. I strongly suggest that the authors should pay more attention to presenting the interaction results between I and Si treatments (rather than the effects of individual treatment) and draw a clear explanation and clear recommendation. 

In the conclusion section, it would be better if the authors will provide economic or cost implications of the findings.

Revision of the Abstract will follow once the comments above are addressed. 

Author Response

Dear Editor,

We would like to thank you and the reviewers for the time and efforts devoted to our manuscript entitled “Physiological, Anatomical, and Agronomic Responses of Cucurbita pepo to Exogenously Sprayed Potassium Silicate at Different Concentrations under Varying Water Regimes” (agronomy-1856718). We have revised the manuscript according to the comments and suggestions pointed out by the reviewers. We have addressed the comments of the reviewers in point-by-point below in red color; in addition, we have highlighted all the associated changes made to the manuscript using track changes.

Yours sincerely,

Authors

Reviewer 2

Overall, I enjoyed reading this well-written manuscript. The results are clearly presented and well discussed. I recommend acceptance of this work after minor revision or addressing a few comments below:

Re: We would like to thank Reviewer 2 for his/her time dedicated to our manuscript. We greatly appreciate his/her positive assessment of our work, encouraging words, and constructive comments for improving our manuscript.

Lines 88 to 90. Would be great to show how much volume of water per hour being employed in Drip Irrigation.

Re: More details have been added as suggested (lines 98-99)

Line 99. Please provide the key standard agronomic practices of the region. “Standard agronomic practices (i.e. xxxxxxxxxxxxx?) were performed”

Re: More details have been added (106-111)

Lines 110-111 Please provide reference to the process being used in the study.

Re: The references for the applied process as well as more details have been added (lines 128-149)

Lines 130-140 Better to provide references for the described processes if possible.

Re: The reference has been added (line 172)

Results:

The Tables and Figures showing the results/data were appropriately presented. I noticed that the results showed significant interactions (Irrigation X Si treatments) in most parameters (Physiological, Anatomical and Agronomic) used in the study. However, this interaction between Irrigation and Si treatments was barely mentioned or emphasized in the results/discussions. I strongly suggest that the authors should pay more attention to presenting the interaction results between I and Si treatments (rather than the effects of individual treatment) and draw a clear explanation and clear recommendation.

Re: The result section has been revised and more details focusing on the interaction effect between the studied factors have been added, particularly the positive impact of potassium silicate under moderate and severe drought stress.

In the conclusion section, it would be better if the authors will provide the economic or cost implications of the findings.

Re: The conclusion has been revised and improved (369-371)

Revision of the Abstract will follow once the comments above are addressed.

Re: The abstract section has been revised and improved

Reviewer 3 Report

This study aimed to observe anatomical and physiological changes in Cucurbita pepo in response to decreasing irrigation rates and increasing rates of Si.

The authors studied many parameters and that is commendable. With that, it was difficult to read through the results.

Main concern was grammar and replication number. Grammar should be revised for clarity, meaning, and ease of reading. Replication number seemed a bit low for a field study.

Comments and suggestions for each section:

Abstract

Would be helpful to spell out acronyms at first use. ~Line 29 - POD, PPO, etc.

Line 31: commas around NPK

Lines 34 and 35: spelling - collenchymatous (check throughout)

Line 36: grammar - exogenously applied (and throughout document)

Double check results match data.

Keywords

Use key words that are not already in the title.

Introduction

Line 61- grammar - exogenously applied

Line 63 - check accepted name of genus. Think Chorisia is now Ceiba.

Use scientific names at first use consistently - Line 64 (maize); Line 68 (wheat)

Materials and Methods

3 replications seems low for a field study. Can you justify?

Need to add general field site conditions - soil type, pH, EC, rainfall, etc.

Line 101 - unit missing (240 ammonium sulfate)

 Line 111 - grammar - commas & ands

Line 113 - grammar - centrifuged

Line 120 - missing temperature degree symbol - 70 C

Line 123 - correct wording and capitalization of equipment (Atomic Absorption....)

Line 131 - spelling - mesophyll

Line 132-133 - spelling - collenchymatous and grammar with use of "and"

Line 135 - unit (um)

Line 144 - have "at" twice

Line 145 - reword for clarity (shoot root, etc.)

Results

I struggled to understand results. It would help to clarify and focus on figures where there is significance. The tables were hard to find the key findings. Tables were very informative, but focusing on main findings and presenting in a way that shows these would be very helpful to the readers.

Grammar needs to be checked throughout. Results reported need to be checked with data shown.

Figure 1 - Add caption for x-axis. In figure caption - name is not written correctly for plant. Suggest using scientific name and cultivar written - cv. Eskandarini or 'Eskandarini'. Include significance explanation.

Section 3.1.3 Mineral Content - Check data reported versus shown. For N and P there did not appear to be a reduction from 75% to 50% ET - it seemed to increase in shoots when comparing those 2 irrigation regimes (going from 75% to 50% ET as stated in text). 

Section 3.2 Anatomical characters - This was the most difficult to understand. There is a table and figures. The table contains very detailed information, but is difficult to understand key points. The Figures 2 and 3 could be interesting and helpful if explained better. The scale does not seem to be the same on the images in Figures 2 and 3. In the caption, it states that the bar=100um, but there is no bar in the images in Figure 2, and there are only 2 images with a bar in Figure 3. This makes it difficult to understand differences. In Figure 2 - periderm and cortex are labeled, so are these the only parameters to look at? In Figure 3 - A and B are not labeled. In Figure 3 caption, suggest explaining C before D.

Conclusion

Double check results match data. Seems to be oversimplifying.

Author Response

Dear Editor,

We would like to thank you and the reviewers for the time and efforts devoted to our manuscript entitled “Physiological, Anatomical, and Agronomic Responses of Cucurbita pepo to Exogenously Sprayed Potassium Silicate at Different Concentrations under Varying Water Regimes” (agronomy-1856718). We have revised the manuscript according to the comments and suggestions pointed out by the reviewers. We have addressed the comments of the reviewers in point-by-point below in red color; in addition, we have highlighted all the associated changes made to the manuscript using track changes.

Yours sincerely,

Authors

Reviewer 3

This study aimed to observe anatomical and physiological changes in Cucurbita pepo in response to decreasing irrigation rates and increasing rates of Si. The authors studied many parameters and that is commendable. With that, it was difficult to read through the results. Main concern was grammar and replication number. Grammar should be revised for clarity, meaning, and ease of reading. Replication number seemed a bit low for a field study.

Re: We would like to thank Reviewer 3 for his/her time dedicated to our manuscript. We greatly appreciate the comment provided by reviewer 3 to improve the quality of the manuscript.  

Comments and suggestions for each section:

Abstract

Would be helpful to spell out acronyms at first use. ~Line 29 - POD, PPO, etc.

Re: The complete terms have been added as suggested (lines 29-30)

Line 31: commas around NPK

Re: Done as suggested (line 32)

Lines 34 and 35: spelling - collenchymatous (check throughout)

Re: The word has been checked throughout the manuscript, thank you for your precision

Line 36: grammar - exogenously applied (and throughout document)

Re: “applied exogenously” has been replaced by “exogenously applied” throughout the manuscript

Keywords

Use keywords that are not already in the title.

Re: The keywords have been modified

Introduction

Line 61- grammar - exogenously applied

Re: “applied exogenously” has been replaced by “exogenously applied” throughout the manuscript

Line 63 - check accepted name of genus. Think Chorisia is now Ceiba.

Re: “Chorisia” has been replaced by “Ceiba” (67)

Use scientific names at first use consistently - Line 64 (maize); Line 68 (wheat)

Re: Done as suggested (lines 70 and 74)

Materials and Methods

3 replications seems low for a field study. Can you justify?

Re: We used three replications as it is common in most agricultural studies

Need to add general field site conditions - soil type, pH, EC, rainfall, etc.

Re: More details have been added as suggested in the text (lines 88-91) as well as the metrological data has been added as supplementary materials (Table S1).

Line 101 - unit missing (240 ammonium sulfate)

Re: The unit has been added (line 111)

 Line 111 - grammar - commas & ands

Re: The sentence has been revised and modified (lines 123-125)

Line 113 - grammar - centrifuged

Re: Done as suggested  (line 126)

Line 120 - missing temperature degree symbol - 70 C

Re: The temperature degree symbol has been added (line 152)

Line 123 - correct wording and capitalization of equipment (Atomic Absorption....)

Re: Done as suggested (line 155)

Line 131 - spelling - mesophyll

Re: Corrected as suggested

Line 132-133 - spelling - collenchymatous and grammar with use of "and"

Re: Revised as suggested (lines 162-166)

Line 145 - reword for clarity (shoot root, etc.)

Re: Revised and modified (line 176)

Results

I struggled to understand results. It would help to clarify and focus on figures where there is significance. The tables were hard to find the key findings. Tables were very informative, but focusing on main findings and presenting in a way that shows these would be very helpful to the readers.

Re: The result section has been revised and more explanations have been added to highlight the interaction effect between the studied factors, particularly the positive impact of potassium silicate under moderate and severe drought stress. The percentages of increase or decrease have been added in the result section please see lines 191-210, 229-237, 257-268, and 290-312 in the revised version.

Grammar needs to be checked throughout. Results reported need to be checked with data shown.

Re: The manuscript has been carefully revised and the results have been checked with the shown data.

Figure 1 - Add caption for x-axis. In figure caption - name is not written correctly for plant. Suggest using scientific name and cultivar written - cv. Eskandarini or 'Eskandarini'. Include significance explanation.

Re: Caption for x-axis has been added and the title has been modified (lines 222-224)

Section 3.2 Anatomical characters - This was the most difficult to understand. There is a table and figures. The table contains very detailed information, but is difficult to understand key points. The Figures 2 and 3 could be interesting and helpful if explained better. The scale does not seem to be the same on the images in Figures 2 and 3. In the caption, it states that the bar=100um, but there is no bar in the images in Figure 2, and there are only 2 images with a bar in Figure 3. This makes it difficult to understand differences. In Figure 2 - periderm and cortex are labeled, so are these the only parameters to look at? In Figure 3 - A and B are not labeled. In Figure 3 caption, suggest explaining C before D.

Re: Figure 2 has been removed and Figure 3 has been improved,  the text has been rewritten and considerably improved.

Double check results match data. Seems to be oversimplifying.

Re: The result section has been carefully checked, and revised as well as more explanations have been added. The percentages of increase or decrease have been added in the results section for all studied parameters for simplifying.

Round 2

Reviewer 1 Report

Authors incorporated all corrections nicely. No further corrections are required.

Author Response

We would like to thank the Reviewer for his time dedicated to our manuscript. We highly appreciate his comments and suggestions in the first round which improved the quality of the manuscript.